# Tessellation-Based Construction of Air Route for Wireless Sensor Networks Employing UAV

**DOI:** 10.3390/s24123867

**Published:** 2024-06-14

**Authors:** CheonWon Choi

**Affiliations:** Department of Computer Engineering, Dankook University, Yongin Si 16890, Republic of Korea; cchoi@dku.edu; Tel.: +82-31-8005-3659

**Keywords:** wireless sensor network, UAV, point of interest, air route, tessellation, Hamiltonian cycle, flight distance

## Abstract

In this paper, we consider a wireless sensor network consisting of an unmanned aerial vehicle (UAV) acting as a sink node and a number of sensor nodes scattered uncertainly on the ground. In the network, the UAV flies to a spatial point called point of interest and hovers to collect environmental data from neighboring sensor nodes. Then, the UAV proceeds to the next point of interest. The UAV must gather data from all the sensor nodes. On the other hand, a shorter round-trip air route of the UAV is more preferred since a battery-operated UAV needs regular recharging. To satisfy the requirement and to adhere to the recommendation as well, especially in the situation where only vague locational information about sensor nodes is available, we propose a scheme that follows three steps. First, it covers the sensor field of the wireless sensor network with three categories of hexagonal tessellations. Secondly, it establishes a point of interest at the centroid of each tile. Thirdly, it constructs an air route of the UAV, which visits every point of interest along a Hamiltonian cycle on the induced graph. Next, we develop a closed-form expression for the exact flight distance attained by the proposed scheme. For comparative evaluation, we discover some optimal schemes that minimize the flight distance by completely inspecting all patterns and corroborating the property of Hamiltonicity. The flight distance along the air route constructed by the proposed scheme is found to be only slightly longer than the flight distance yielded by an optimal scheme. Furthermore, the proposed scheme is proven to be practically valid when a common multicopter is employed as the sink node.

## 1. Introduction

A wireless sensor network consists of sink nodes and sensor nodes. In the network, a sensor node amasses environmental information in the vicinity and wirelessly delivers it to a sink node [1]. Accordingly, a sink node collects data from neighboring sensor nodes. Wireless sensor networks have been and are expected to be deployed to various areas, often in harsh environments. For example, a wireless sensor network is used to construct a large database of environmental observations for weather forecasting [2]. A wireless sensor network is also deployed for herd management, assessing the condition of animals and reporting the data to the farm manager [3]. A wireless sensor network is further deployed to a region that is difficult to access due to extreme conditions such as high temperatures, high humidity, and high pressure [4].

Meanwhile, unmanned aerial vehicle (UAV) technologies have spread widely and become popular in various fields, including radar localization, wildfire management, agricultural monitoring, border surveillance, meteorological monitoring, and rescue missions [5]. The benefits brought by the use of UAVs, for example, flexibility in constructing a network, effectiveness in saving energy, and safety for humans, exhort wireless sensor networks to increasingly employ UAVs for collecting data from sensor nodes on the ground [5,6].

In this paper, we consider a wireless sensor network as follows. In the wireless sensor network, sensor nodes are scattered on the ground. As usual, these sensor nodes amass environmental information in the vicinity of them. However, the geographical distribution of sensor nodes is not precisely known, and rough information about the locations of sensor nodes, e.g., indistinct boundary of the region in which sensor nodes sojourn, is only available to us (Such a situation can happen due to a harsh environment in which the wireless sensor network is laid or irregular mobilities of sensor nodes comprising the wireless sensor network). To collect data from sensor nodes, the wireless sensor network employs a single UAV as a sink node. In order to support the UAV in flying over the wireless sensor network, a number of spatial points called points of interest are then established. Also, an air route is constructed for the UAV to visit these points of interest. While flying along the air route, the UAV hovers whenever it arrives at each point of interest. Then, according to a medium access control (MAC) scheme, sensor nodes near the point of interest send the environmental information that they have amassed in the vicinity of them to the lingering UAV. In such a way, the UAV, which acts as a sink node, is able to gather data from sensor nodes.

There have been several studies on the establishment of points of interest and the construction of air routes in a wireless sensor network in which a UAV plays the role of a sink node [4,5,6,7,8,9,10,11,12,13,14,15,16,17,18,19]. In [7], heavily loaded sensor nodes were designated as special sensor nodes that a mobile sink would visit. Then, a path for the mobile sink was formed to include only the special sensor nodes. In [9], a grid of geographical points was assumed over a wireless sensor network. Then, some points that a drone would visit were selected, and a path to pass selected points was created so that the sum of flying time and hovering time was minimized. In [12], geographical points at which a mobile sink collects data were built, and a path that passes these points was designed so that the cost incurred by the mobile sink’s flying between any two points as well as the cost brought by the sensor node’s forwarding data to a point. In [16], sensor nodes were clustered in a wireless sensor network. Then, the head of each cluster that a UAV visits was selected so as to maximize the amount of efficiently transmitted data. In [19], a digital twin was created to determine the path for UAVs. A best path was empirically determined based on the experience of the digital twin in a cyberspace that resembles the real environment. Then, information about the best path was delivered to UAVs. The studies as listed above suggested distinct ways of establishing points of interest and constructing an air route for the UAV. However, most of them were commonly carried out under the assumption that considerable information about the locations of sensor nodes was available.

In the wireless sensor network under discussion, points of interest should be configured so that the UAV, which acts as a sink node, is capable of collecting data from all the sensor nodes in principle. On the other hand, it is more preferred to construct a shorter round-trip air route for the UAV in the wireless sensor network since a UAV is typically powered by a battery, which should be recharged or replaced regularly. To satisfy such a requirement and to follow such a recommendation as well, especially in the situation that scanty locational information about sensor nodes is only available to us, we propose a scheme as follows. First, we employ three categories of hexagonal tessellations [20], which bear the property of Hamiltonicity [21] underneath, to cover the sensor field of the wireless sensor network. Next, we establish a point of interest at the center of each tile belonging to the adopted tessellation so that the UAV is definitely able to collect data from all the sensor nodes in the wireless sensor network. Finally, we construct a round-trip air route along a Hamiltonian cycle, which is sought on the employed tessellation, so that the UAV can visit every point of interest while the flight distance, i.e., the length of the air route, is reduced. Then, we develop a closed-form expression of the exact flight distance attained by the proposed scheme. For comparative evaluation, we also discover an optimal scheme that minimizes the flight distance, for some radii of the sensor field by completely inspecting all patterns and corroborating the property of Hamiltonicity. Then, the proposed scheme is compared with the optimal scheme for flight distance. In addition, the proposed scheme is evaluated in the situation where a common multicopter is employed as a sink node in the wireless sensor network.

The main contributions of this paper are as follows:We build three categories of hexagonal tessellations that bear the property of Hamiltonicity underneath.To build an air route for the UAV, we propose a scheme which covers the sensor field with a hexagonal tessellation belonging to one of the three categories, establishes a point of interest at the centroid of each tile, and constructs an air route along a Hamiltonian cycle embedded on the hexagonal tessellation.We present a closed-form expression of the flight distance yielded by the proposed scheme.We find a suboptimal scheme that minimizes the number of points of interest by completely inspecting all patterns. Also, we discover an optimal scheme that minimizes the flight distance by corroborating the Hamiltonicity in suboptimal tessellation.We provide a universal lower bound on the flight distance in a closed form.

The paper is organized as follows. In Section 2, we describe the wireless sensor network in which a UAV plays the role of a sink node. In Section 3, we present a scheme that covers the sensor field of the wireless sensor network with a hexagonal tessellation, establishes a point of interest at the centroid of each hexagonal tile, and constructs a round-trip air route of the UAV along a Hamiltonian cycle embedded in the hexagonal tessellation. In Section 4, we develop a close-form expression of the exact flight distance attained by the proposed scheme. Also, we obtain the flight distance yielded by a scheme, which is optimal in the sense of minimizing the flight distance, by completely inspecting all patterns and corroborating the property of Hamiltonicity as well. Further, we derive a simple lower bound on the flight distance. Section 5 is devoted to numerical examples that comparatively evaluate the proposed scheme.

## 2. Wireless Sensor Network Employing UAV

In this paper, we consider a wireless sensor network consisting of a UAV and a number of sensor nodes. The UAV, which plays the role of a sink node, flies over the sensor nodes and receives data from them. On the other hand, sensor nodes, which are randomly scattered on the ground, gather environmental data in the vicinity and wirelessly send the data to the UAV.

In the wireless sensor network, the geographical distribution of the sensor nodes is not precisely known, and only marginal information about the locations of sensor nodes is available to us. Assume that only a vague region that encloses all the sensor nodes is known to us in the wireless sensor network under consideration. In addition, assume that the vague region takes the shape of a circular disk with radius α. Define the sensor field of a wireless sensor network to be a region in a geometric shape that encloses all the sensor nodes in the network. Then, we set the vague region to be the sensor field of the wireless sensor network (See Figure 1). Ideally, the sensor field of the wireless sensor network can be regarded as a circular disk whose boundary forms the circumcircle of all the sensor nodes comprising the wireless sensor network. Note that sensor fields can be defined or modeled to have a variety of shapes. For example, a sensor field can be defined as being in the shape of a polygon in which all the sensor nodes are inscribed. Since vague information about the geographic distribution of sensor nodes is assumed to be available, it is appropriate to model the sensor field as a circular disk, which is one of the most general shapes.

The UAV acting as a sink node flies over the sensor field to receive data from the sensor nodes lying on the sensor field. To help the UAV efficiently navigate the sensor field for collecting data, several spatial points, which are called points of interest, are established on the sensor field (See Figure 2). Then, a round-trip air route for the UAV is constructed so that the UAV is able to visit every point of interest (See Figure 3).

The UAV flies along the pre-determined air route and visits the points of interest. As the UAV arrives at a point of interest, it hovers and receives data from the sensor nodes, which reside in a region called the coverage associated with the point of interest (See Figure 2). Note that the coverage associated with a point of interest is determined by many factors, for example, the transmission power of a sensor node, altitude of the UAV, and beamwidth of the antenna at the UAV. We set that the coverage associated with each point of interest identically takes the shape of a circular disk with radius β (See Figure 3). Such a setting can be realized by assuming that the sensor field is flat and the UAV flies at a fixed altitude. In practice, the terrain on which a wireless sensor network is deployed can be mountainous. Then, coverages can have different sizes, even if the UAV navigates at a fixed altitude. In such a case, we assume that each coverage has the minimum among the distinct sizes.

As the UAV finishes collecting data from the sensor nodes sojourning in the coverage associated with a point of interest, the UAV leaves the point of interest and flies to the next point of interest along the air route.

## 3. Tessellation-Based Establishment of Points of Interest and Construction of Air Route

Consider a wireless sensor network employing a UAV, as depicted in Section 2. For the UAV to collect data from the sensor nodes in the network, several points of interest are established on the sensor field, and an air route is also constructed along the points of interest a priori. As addressed in the introduction, points of interest should be established so that the UAV is able to collect data from all the sensor nodes in the wireless sensor network. Furthermore, it is more preferred to construct a shorter round-trip air route along which the UAV flies. To satisfy the requirement and to follow the recommendation as well, we propose a scheme, denoted by $\sigma^{*}$, which employs a hexagonal tessellation [20] to cover the sensor field of the wireless sensor network, establishes a point of interest at the centroid of each hexagonal tile, and constructs a round-trip air route of the UAV by finding a Hamiltonian cycle [21].

### 3.1. Tessellation-Based Establishment of Points of Interest

A tessellation is a collection of tiles in polygonal shapes that are mutually attached without gaps or overlaps [20]. Figure 4 shows a tessellation, which consists of tiles in two polygonal shapes.

In order to establish points of interest in the situation where vague geographical information about the sensor nodes is only available, we consider using a tessellation to cover the sensor field of the wireless sensor network and then setting up a point of interest per tile that comprises the tessellation.

There are a variety of candidate tessellations for covering the sensor field. Remind the requirement that points of interest should be established so that the UAV is able to gather data from all the sensor nodes lying in the wireless sensor network. Also, recall the recommendation that a shorter round-trip air route along which the UAV visits all the points of interest is more preferred. Thus, to cover the sensor field, a candidate tessellation is definitely favored if it consists of a small number of tiles, its tiles take appropriate shapes, and the area of each tile is as large as possible. Hereafter, while bearing the requirement and recommendation in mind, we will gradually narrow candidate tessellations for covering the sensor field down to three classes of hexagonal tessellations, i.e., tessellations consisting of regular hexagons [20]. Then, we propose using these classes of hexagonal tessellations to cover the sensor field and establishing a point of interest at the centroid of each tile.

Tessellations can be divided into monohedral tessellations, i.e., tessellations with tiles in the same polygonal shape, and non-monohedral tessellations [20]. Figure 5 illustrates a monohedral tessellation, which consists of triangular tiles. Also, note that the tessellation in Figure 4 is an example of non-monohedral tessellation.

As the sensor field is covered by a tessellation, a point of interest is established in the convex hull of each tile belonging to the tessellation. Then, the coverage associated with the point of interest, which takes the shape of a circular disk with radius β, is also built such that the point of interest is centered on the coverage. For the UAV to receive data from all the sensor nodes, each tile of the tessellation should be enclosed in the corresponding coverage. Further, to reduce the number of points of interest, it is advantageous that a coverage forms the smallest enclosing disk of the corresponding tile. A non-monohedral tessellation consists of tiles in two or more distinct shapes. In case that the sensor node is covered by such a tessellation, the coverage corresponding to a tile cannot be the smallest enclosing disk as far as the tile is not the largest tile in the tessellation. As a result, the sensor field can be covered by a monohedral tessellation, which consists of a smaller number of tiles than a non-monohedral tessellation. From now on, we thus confine our attention to monohedral tessellations for candidate tessellations to cover the sensor field.

Monohedral tessellations can be categorized into edge-to-edge monohedral tessellations, i.e., monohedral tessellations in which each tile only shares one full side with any adjacent tile, and non-edge-to-edge monohedral tessellations [20]. Figure 6 exhibits a non-edge-to-edge monohedral tessellation consisting of rectangular tiles. The tessellation shown in Figure 5 is a typical edge-to-edge monohedral tessellation.

To reduce the number of points of interest, it is more preferred to employ a tessellation in which tiles are positioned so that coverages are less overlapped. Henceforth, we thus concentrate on edge-to-edge monohedral tessellations for candidate tessellations to cover the sensor field.

A tile can take the shape of either a convex polygon or a concave polygon. Figure 7 demonstrates an edge-to-edge monohedral tessellation comprised of tiles that take the shape of a concave polygon. Note that the tessellation exhibited in Figure 5 is an edge-to-edge monohedral tessellation consisting of tiles that take the shape of a convex polygon.

For any concave tile that is enclosed in a coverage, note that there is a larger convex tile that can be enclosed in the same coverage. Since a larger tile is more favored to reduce the number of points of interest, we focus on edge-to-edge monohedral tessellations with convex tiles hereafter.

Note that tiles in the shape of any triangle can comprise an edge-to-edge monohedral tessellation. Also, tiles in the shape of any convex quadrilateral can form an edge-to-edge monohedral tessellation. On the other hand, 8 types of convex pentagons and 3 types of convex hexagons are known to be able to make up edge-to-edge monohedral tessellations [20,22]. However, there is no edge-to-edge monohedral tessellation that consists of tiles in the shape of a convex polygon with more than 6 sides [20]. Recall that a larger tile is more suitable for reducing the number of points of interest. Among the candidate polygons that can be used for edge-to-edge monohedral tessellations, note that an equilateral equiangular hexagon is the largest polygon that can be enclosed in a coverage that takes the shape of a circular disk. Henceforward, we only consider hexagonal tessellations, i.e., edge-to-edge monohedral tessellations in which tiles are in the shape of a regular (equilateral and equiangular) hexagon [20], for candidate tessellations to cover the sensor field. Note that a candidate hexagonal tessellation consists of hexagonal tiles whose sides are β in length, i.e., hexagonal tiles inscribed in coverages (See Figure 8).

Since a tessellation is more favored if it consists of a smaller number of tiles, we may use a hexagonal tessellation that covers the sensor field with the minimum number of tiles. However, searching for such a hexagonal tessellation is not tractable, especially when the sensor field is relatively large. Furthermore, a minimum number of points of interest does not necessarily invoke the shortest round-trip air route of a UAV. Thus, we would rather confine our attention to three classes of hexagonal tessellations that possess a useful property called Hamiltonicity [21], as follows.

Consider the tiles at the boundary of a hexagonal tessellation. Connect the centroids of each pair of adjacent tiles using a straight-line segment. Then, a polygon can be induced by the tessellation. Recall that the coverage associated with a point of interest takes the shape of a circular disk with radius β. For n∈N, let $T_{n}^{\left(1\right)}$ denote a hexagonal tessellation that induces a regular hexagon whose sides are $\sqrt{3}n\beta$ in length. (Hereafter, we call $T_{n}^{\left(1\right)}$ the hexagonal tessellation of category 1 and scale n.) For n∈N, let $T_{n}^{\left(2\right)}$ and $T_{n}^{\left(3\right)}$ denote tessellations that induce distinct isogonal hexagons. In the hexagon induced by tessellation $T_{n}^{\left(2\right)}$, the lengths of 3 sides are commonly $\sqrt{3}n\beta$ while the lengths of the other 3 sides are equal to $\sqrt{3}(n+1)\beta$. On the other hand, tessellation $T_{n}^{\left(3\right)}$ induces the hexagon in which 3 sides are commonly $\sqrt{3}n$ and the other 3 sides are identically $\sqrt{3}(n+2)$ in length (Henceforth, we call $T_{n}^{\left(2\right)}$ ($T_{n}^{\left(3\right)}$) the hexagonal tessellation of categories 2 (category 3) and scale n). Figure 9 illustrates tessellations $T_{2}^{\left(1\right)}$, $T_{2}^{\left(2\right)}$ and $T_{2}^{\left(3\right)}$.

In addition, let $T_{0}^{\left(1\right)}$, $T_{0}^{\left(2\right)}$, and $T_{0}^{\left(3\right)}$ denote hexagonal tessellations with 1, 3, and 6 hexagonal tiles, respectively, as illustrated in Figure 10.

From now on, we confine our attention to three categories of tessellations $\left\{T_{n}^{\left(1\right)}:n\in \{0\}\cup N\right\}$, $\left\{T_{n}^{\left(2\right)}:n\in \{0\}\cup N\right\}$ and $\left\{T_{n}^{\left(3\right)}:n\in \{0\}\cup N\right\}$ to cover the sensor field of the wireless sensor network.

The following theorems state some properties exhibited by three categories of hexagonal tessellations when they are used to cover the sensor field. (Later, these properties will be used to derive the flight distance, i.e., the length of a round-trip air route.) First, the theorem below shows the number of tiles which comprise each of $T_{n}^{\left(1\right)}$, $T_{n}^{\left(2\right)}$ and $T_{n}^{\left(3\right)}$, or equivalently, the number of points of interest established when the sensor field is covered by each of them.

**Theorem** **1.***Suppose that the sensor field is covered by tessellation* $T_{n}^{\left(c\right)}$ *for* ∈{1,2,3} *and* n∈{0}∪N. *Let* $K_{n}^{\left(c\right)}$ *denote the number of points of interest established when tessellation* $T_{n}^{\left(c\right)}$ *is used to cover the sensor field for* c∈{1,2,3} *and* n∈{0}∪N. *Then,*(1)$$\begin{array}{lll} K_{n}^{\left(1\right)} & = & 3n^{2}+3n+1\\ K_{n}^{\left(2\right)} & = & 3n^{2}+6n+3\\ K_{n}^{\left(3\right)} & = & 3n^{2}+9n+6 \end{array}$$*for* n∈{0}∪N.

**Proof.** A proof of Theorem 1 is given in Appendix A. □

Secondly, the following theorem reveals the maximum size of the sensor field that can be enclosed in each of $T_{n}^{\left(1\right)}$, $T_{n}^{\left(2\right)}$ and $T_{n}^{\left(3\right)}$.

**Theorem** **2.***For* c∈{1,2,3} *and* n∈{0}∪N, *let* $R_{n}^{\left(c\right)}$ *denote the maximum radius of the sensor field that can be enclosed in tessellation* $T_{n}^{\left(c\right)}$. *Then,*(2)$$\begin{array}{lll} R_{0}^{\left(1\right)} & = & \frac{\sqrt{3}}{2}\cdot \beta \\ R_{0}^{\left(2\right)} & = & \beta \\ R_{0}^{\left(3\right)} & = & \sqrt{3}\cdot \beta \end{array}\;\begin{array}{lll} R_{2m-1}^{\left(1\right)} & = & (3m-1)\cdot \beta \\ R_{2m-1}^{\left(2\right)} & = & \sqrt{9m^{2}-3m+1}\cdot \beta \\ R_{2m-1}^{\left(3\right)} & = & 3m\cdot \beta \end{array}\;\begin{array}{lll} R_{2m}^{\left(1\right)} & = & \sqrt{9m^{2}+3m+1}\cdot \beta \\ R_{2m}^{\left(2\right)} & = & (3m+1)\cdot \beta \\ R_{2m}^{\left(3\right)} & = & \sqrt{9m^{2}+9m+3}\cdot \beta \end{array}$$*for* m∈N.

**Proof.** A proof of Theorem 2 is given in Appendix B. □

From Theorems 1 and 2, we can obtain an inequality among the numbers of points of interest when the sensor field is covered by hexagonal tessellations belonging to the three categories, respectively. Also, we can derive the relation between the maximum radii of the sensor field that can be enclosed in hexagonal tessellations belonging to the three categories, respectively, as presented in the corollary below.

**Corollary** **1.***For* n∈{0}∪N*,*(3)$$K_{n}^{\left(1\right)}\leq K_{n}^{\left(2\right)}\leq K_{n}^{\left(3\right)}$$(4)$$R_{n}^{\left(1\right)}\leq R_{n}^{\left(2\right)}\leq R_{n}^{\left(3\right)}.$$

**Proof.** A proof of Corollary 1 is given in Appendix C. □

Suppose that we cover the sensor field of radius α by a tessellation which belongs to one of the three categories of hexagonal tessellations. Consider the hexagonal tessellation of category $\Psi_{\alpha }$ and scale $\Lambda_{\alpha }$, where $\Psi_{\alpha }\in \left\{1,2,3\right\}$ and $\Lambda_{\alpha }\in \{0\}\cup N$ uniquely satisfy
(5)$$K_{\Lambda_{\alpha }}^{\left(\Psi_{\alpha }\right)}=min\left\{K_{n}^{\left(c\right)}:R_{n}^{\left(c\right)}\geq \alpha \right.\left.for\;c\in \left\{1,2,3\right\}\;and\;n\in \{0\}\cup N\right\}$$
for each α∈(0,∞). Then, we propose scheme $\sigma^{*}$ that uses tessellation $T_{\Lambda_{\alpha }}^{\left(\Psi_{\alpha }\right)}$ to cover the sensor field and establishes a point of interest at the centroid of each tile comprising tessellation $T_{\Lambda_{\alpha }}^{\left(\Psi_{\alpha }\right)}$.

### 3.2. Tessellation-Based Construction of Air Route

In this section, we propose a scheme which constructs a round-trip air route of the UAV for collecting data over the sensor field of the wireless sensor network.

For category c∈{1,2,3} and scale n∈{0}∪N, we can construct a graph induced by tessellation $T_{n}^{\left(c\right)}$ as follows. Let a node represent a tile of $T_{n}^{\left(c\right)}$. Let a link connect a pair of nodes if two tiles corresponding to the nodes are adjacent. Then, such sets of nodes and links form the graph induced by tessellation $T_{n}^{\left(c\right)}$. For c∈{1,2,3} and n∈{0}∪N, let $G_{n}^{\left(c\right)}$ denote the graph induced by tessellation $T_{n}^{\left(c\right)}$. Figure 11 demonstrates graphs $G_{2}^{\left(1\right)}$, $G_{2}^{\left(2\right)}$ and $G_{2}^{\left(3\right)}$, which are induced by tessellations $T_{2}^{\left(1\right)}$, $T_{2}^{\left(2\right)}$ and $T_{2}^{\left(3\right)}$, respectively. Note that graph $G_{n}^{\left(c\right)}$ is a triangular grid graph for all c∈{1,2,3} and n∈{0}∪N [23].

In a graph, a path is a sequence of distinct nodes in which there is a link between two consecutive nodes. In a graph, a cycle is a special path in which the starting node is identical to the ending node. A cycle in a graph is called Hamiltonian if the cycle visits every node in the graph. A graph is said to be Hamiltonian if a Hamiltonian cycle exists in the graph [21]. The theorem below shows that graphs $G_{n}^{\left(1\right)}$, $G_{n}^{\left(2\right)}$ and $G_{n}^{\left(3\right)}$ are Hamiltonian.

**Theorem** **3.***Graphs* $G_{n}^{\left(1\right)}$, *$G_{n}^{\left(2\right)}$**and $G_{n}^{\left(3\right)}$**are Hamiltonian for all n∈{0}∪N*.

**Proof.** A proof of Theorem 3 is given in Appendix D. □

A Hamiltonian cycle in $G_{n}^{\left(1\right)}$ is not necessarily unique. Figure 12 illustrates exemplary Hamiltonian cycles in graphs $G_{2}^{\left(1\right)}$, $G_{2}^{\left(2\right)}$ and $G_{2}^{\left(3\right)}$.

Recall category $\Psi_{\alpha }$ and scale $\Lambda_{\alpha }$ defined in (5). Suppose that we cover the sensor field with tessellation $T_{\Lambda_{\alpha }}^{\left(\Psi_{\alpha }\right)}$. Note that tessellation $T_{\Lambda_{\alpha }}^{\left(\Psi_{\alpha }\right)}$ consists of $K_{\Lambda_{\alpha }}^{\left(\Psi_{\alpha }\right)}$ tiles. Thus, graph $G_{\Lambda_{\alpha }}^{\left(\Psi_{\alpha }\right)}$ which is induced by tessellation $T_{\Lambda_{\alpha }}^{\left(\Psi_{\alpha }\right)}$ is comprised by $K_{\Lambda_{\alpha }}^{\left(\Psi_{\alpha }\right)}$ nodes. Let $\left\{v_{1},\cdots ,v_{K_{\Lambda_{\alpha }}^{\left(\Psi_{\alpha }\right)}}\right\}$ denote the set of nodes in graph $G_{\Lambda_{\alpha }}^{\left(\Psi_{\alpha }\right)}$. From Theorem 4, there is a Hamiltonian cycle in graph $G_{\Lambda_{\alpha }}^{\left(\Psi_{\alpha }\right)}$. Let $\left\{v_{\pi_{1}},\cdots ,v_{\pi_{K_{\Lambda_{\alpha }}^{\left(\Psi_{\alpha }\right)}}},v_{\pi_{1}}\right\}$ denote a Hamiltonian cycle in graph $G_{\Lambda_{\alpha }}^{\left(\Psi_{\alpha }\right)}$, where $\left\{\pi_{1},\cdots ,\pi_{K_{\Lambda_{\alpha }}^{\left(\Psi_{\alpha }\right)}}\right\}$ is a permutation of $\left\{1,\cdots ,K_{\Lambda_{\alpha }}^{\left(\Psi_{\alpha }\right)}\right\}$. For $j\in \{1,\cdots ,K_{\Lambda_{\alpha }}^{\left(\Psi_{\alpha }\right)}\}$, let $P_{j}$ denote the spatial location, e.g., triple of longitude, latitude and altitude, of the point of interest which lies at the centroid of the tile corresponding to node $v_{j}$. Then, we propose scheme $\sigma^{*}$ that constructs a round-trip air route that starts from $P_{\pi_{1}}$ and also ends at $P_{\pi_{1}}$ as follows. The air route sequentially visits $P_{\pi_{j}}$ for $j\in \{1,\cdots ,K_{\Lambda_{\alpha }}^{\left(\Psi_{\alpha }\right)}\}$. From $P_{\pi_{j}}$ to $P_{\pi_{j+1}}$, the air route is built as a straight-line segment for $j\in \{1,\cdots ,K_{\Lambda_{\alpha }}^{\left(\Psi_{\alpha }\right)}-1\}$. From $P_{\pi_{K_{\Lambda_{\alpha }}^{\left(\Psi_{\alpha }\right)}}}$ to $P_{\pi_{1}}$, the air route is also constructed as a straight-line segment.

For example, suppose that radius of the sensor field α is equal to $\sqrt{3}\beta$. Then, category $\Psi_{\alpha }$ and scale $\Lambda_{\alpha }$ are calculated to be 3 and 0, respectively. Thus, tessellation $T_{0}^{\left(3\right)}$ is used to cover the sensor field according to scheme $\sigma^{*}$. Then, a point of interest is established at the centroid of each tile in tessellation $T_{0}^{\left(3\right)}$ (See Figure 13a). Next, graph $G_{0}^{\left(3\right)}$ which consists of 6 nodes, denoted by $v_{1},\cdots ,v_{6}$, is induced from tessellation $T_{0}^{\left(3\right)}$ as shown in Figure 13b. Consider sequence $\left\{1,2,4,5,6,3\right\}$, which is a permutation of $\left\{1,2,3,4,5,6\right\}$. Then, $\left\{v_{1},v_{2},v_{4},v_{5},v_{6},v_{3},v_{1}\right\}$ is a Hamiltonian cycle on induced graph $G_{0}^{\left(3\right)}$. (See Figure 13c). Finally, a round-trip air route, in which every point of interest is constructed along the Hamiltonian cycle over the sensor field, is shown in Figure 13d.

## 4. Analysis of Flight Distance

In Section 3, we proposed scheme $\sigma^{*}$ for establishing points of interest over the sensor field and building a round-trip air route that visits every point of interest. In this section, we derive the flight distance attained by scheme $\sigma^{*}$, i.e., the length of the round-trip air route constructed according to scheme $\sigma^{*}$. For the comparative evaluation of proposed scheme $\sigma^{*}$, which will be carried out in the following section, we also consider a family of schemes that use hexagonal tessellations and analyze the flight distances achieved by such schemes as well.

In the wireless sensor network under discussion, recall that the sensor field takes the shape of circular disk with radius α and each coverage is in the shape of a circular disk with radius β. Consider a family of schemes as follows. To cover the sensor field, each scheme belonging to the family uses a hexagonal tessellation in which each hexagonal tile is inscribed in the corresponding coverage, i.e., every side of each tile is β in length. Then, the scheme establishes a point of interest at the centroid of each tile. At last, the scheme constructs the shortest round-trip air route to visit every point of interest.

For i∈N, consider a set of distinct hexagonal tessellations, each of which consists of i hexagonal tiles with sides of length β. For i∈N, let $\nu_{i}$ denote the number of such distinct hexagonal tessellations. Note that $\nu_{1}=\nu_{2}=1$ and $\nu_{3}=3$ for example. For size i∈N, let $\left\{S_{i}^{\left[p\right]}:p\in \{1,\cdots ,\nu_{i}\}\right\}$ denote the set of such distinct hexagonal tessellations. (Hereafter, we call $S_{i}^{\left[p\right]}$ hexagonal tessellation of pattern p and size i). For size i∈N and pattern $p\in \{1,\cdots ,\nu_{i}\}$, let $\sigma_{i}^{\left[p\right]}$ denote the scheme which uses tessellation $S_{i}^{\left[p\right]}$ to cover the sensor field. Then, $\left\{\sigma_{i}^{\left[p\right]}:p\in \left\{1,\cdots ,\nu_{i}\right\},i\in N\right\}$ represents the family of schemes that we focus on.

For size i∈N and pattern $p\in \{1,\cdots ,\nu_{i}\}$, let $Q_{i}^{\left[p\right]}$ denote the maximum radius of the sensor field that can be enclosed in tessellation $S_{i}^{\left[p\right]}$. For size i∈N and pattern $p\in \{1,\cdots ,\nu_{i}\}$, let $N(\alpha ,\sigma_{i}^{\left[p\right]})$ denote the number of points of interest established over the sensor field of radius α by scheme $\sigma_{i}^{\left[p\right]}$ if the sensor field can be covered by tessellation $S_{i}^{\left[p\right]}$. Then,
(6)$$N\left(\alpha ,\sigma_{i}^{\left[p\right]}\right)=i$$
if $Q_{i}^{\left[p\right]}\geq \alpha$. Otherwise, we set $N\left(\alpha ,\sigma_{i}^{\left[p\right]}\right)=\infty$. Also, for size i∈N and pattern $p\in \left\{1,\cdots ,\nu_{i}\right\},$ let $D(\alpha ,\sigma_{i}^{\left[p\right]})$ represent the flight distance attained by scheme $\sigma_{i}^{\left[p\right]}$, i.e., the length of the round-trip air route that visits every point of interest established over the sensor field of radius α according to scheme $\sigma_{i}^{\left[p\right]}$ if the sensor field can be covered by tessellation $S_{i}^{\left[p\right]}$. We set $D\left(\alpha ,\sigma_{i}^{\left[p\right]}\right)=\infty$ if $Q_{i}^{\left[p\right]}<\alpha$.

First, recall category $\Psi_{\alpha }$ and scale $\Lambda_{\alpha }$ defined in (5). The theorem below shows the flight distance attained by proposed scheme $\sigma^{*}$, i.e., the length of a round-trip air route when the sensor field is covered by tessellation $T_{\Lambda_{\alpha }}^{\left(\Psi_{\alpha }\right)}$.

**Theorem** **4.***Suppose that we employ scheme* $\sigma^{*}$ *to establish points of interest over* *the sensor field of radius α* *and construct a round-trip air route that visits every point of interest. Then,*(7)$$D\left(\alpha ,\sigma^{*}\right)=\sqrt{3}\beta K_{\Lambda_{\alpha }}^{\left(\Psi_{\alpha }\right)}$$*where* $K_{\Lambda_{\alpha }}^{\left(\Psi_{\alpha }\right)}$ *is the number of points of interest established over the sensor field of radius* α *according to scheme* $\sigma^{*}$.

**Proof.** A proof of Theorem 4 is given in Appendix E. □

Secondly, consider a set of hexagonal tessellations $\left\{S_{i}^{\left[p\right]}:p\in \left\{1,\cdots ,\nu_{i}\right\},i\in N\right\}$. Focus on scheme $\sigma_{i}^{\left[p\right]}$ which employs tessellation $S_{i}^{\left[p\right]}$ for pattern $p\in \left\{1,\cdots ,\nu_{i}\right\}$ and size i∈N. Given the radius of the sensor field α∈(0,∞), let $\Xi_{\alpha }$ and $\Delta_{\alpha }$ denote pattern and size that satisfy
(8)$$D(\alpha ,\sigma_{\Delta_{\alpha }}^{\left[\Xi_{\alpha }\right]})=min\left\{D(\alpha ,\sigma_{i}^{\left[p\right]}):Q_{i}^{\left[p\right]}\geq \alpha \right.\left.for\;p\in \left\{1,\cdots ,\nu_{i}\right\}\;and\;i\in N\right\}$$
Then, scheme $\sigma_{\Delta_{\alpha }}^{\left[\Xi_{\alpha }\right]}$ invokes the minimum flight distance when the radius of the sensor field is α. Henceforth, we call scheme $\sigma_{\Delta_{\alpha }}^{\left[\Xi_{\alpha }\right]}$ an optimal scheme in the sense of minimizing the flight distance. Also, let $\sigma^{†}$ denote a scheme that uses scheme $\sigma_{\Delta_{\alpha }}^{\left[\Xi_{\alpha }\right]}$ for radius α∈(0,∞) collectively. Note that optimal scheme $\sigma_{\Delta_{\alpha }}^{\left[\Xi_{\alpha }\right]}$ is hard to find in general.

Consider set of hexagonal tessellations $\left\{S_{i}^{\left[p\right]}:p\in \{1,\cdots ,\nu_{i}\}\right\}$ for size i∈N. Recall that $Q_{i}^{\left[p\right]}$ denotes the maximum radius of the sensor field that can be enclosed in tessellation $S_{i}^{\left[p\right]}$ for $p\in \{1,\cdots ,\nu_{i}\}$. Let $\Phi_{i}$ denote a pattern such that
(9)$$Q_{i}^{\left[\Phi_{i}\right]}=max\left\{Q_{i}^{\left[p\right]}:p\in \{1,\cdots ,\nu_{i}\}\right\}$$
for size i∈N. Then, scheme $\sigma_{i}^{\left[\Phi_{i}\right]}$, which employs tessellation $Q_{i}^{\left[\Phi_{i}\right]}$, satisfies
(10)$$N\left(\alpha ,\sigma_{i}^{\left[\Phi_{i}\right]}\right)=min\left\{N\left(\alpha ,\sigma_{j}^{\left[p\right]}\right):p\in \left\{1,\cdots ,\nu_{i}\right\},j\in \{i,i+1,\cdots \}\right\}$$
for $\alpha \in (Q_{i-1}^{\left[\Phi_{i-1}\right]},Q_{i}^{\left[\Phi_{i}\right]}]$, i.e., scheme $\sigma_{i}^{\left[\Phi_{i}\right]}$ invokes the minimum number of points of interest among the schemes which use tessellations that can enclose the sensor field of radius $\alpha \in (Q_{i-1}^{\left[\Phi_{i-1}\right]},Q_{i}^{\left[\Phi_{i}\right]}]$. However, it is not guaranteed that
(11)$$D\left(\alpha ,\sigma_{i}^{\left[\Phi_{i}\right]}\right)=min\left\{D\left(\alpha ,\sigma_{j}^{\left[p\right]}\right):p\in \left\{1,\cdots ,\nu_{i}\right\},j\in \{i,i+1,\cdots \}\right\}$$
for $\alpha \in (Q_{i-1}^{\left[\Phi_{i-1}\right]},Q_{i}^{\left[\Phi_{i}\right]}]$. Thus, scheme $\sigma_{i}^{\left[\Phi_{i}\right]}$ is not necessarily an optimal scheme. Hereafter, we call $\sigma_{i}^{\left[\Phi_{i}\right]}$ a suboptimal scheme in the sense of minimizing the number of points of interest. Also, let $\sigma^{‡}$ denote a scheme that uses suboptimal scheme $\sigma_{i}^{\left[\Phi_{i}\right]}$ for radius $\alpha \in (Q_{i-1}^{\left[\Phi_{i-1}\right]},Q_{i}^{\left[\Phi_{i}\right]}]$ and size i∈N collectively.

**Theorem** **5.***For radius* $\alpha \in (Q_{i-1}^{\left[\Phi_{i-1}\right]},Q_{i}^{\left[\Phi_{i}\right]}]$ *and size* i∈N, *suboptimal scheme* $\sigma_{i}^{\left[\Phi_{i}\right]}$ *is optimal scheme is identical to optimal scheme* $\sigma_{\Delta_{\alpha }}^{\left[\Xi_{\alpha }\right]}$ *if the graph induced by tessellation* $S_{i}^{\left[\Phi_{i}\right]}$ *is Hamiltonian.*

**Proof.** Tessellation $S_{i}^{\left[\Phi_{i}\right]}$ consists of i tiles. Also, the straight-line distance between two adjacent points of interest is equal to $\sqrt{3}\beta$. Thus, the flight distance of a round-trip air route must be greater than or equal to $\sqrt{3}\beta i$. If the graph induced by tessellation $S_{i}^{\left[\Phi_{i}\right]}$ is Hamiltonian, there is a Hamiltonian cycle, which consists of i links. If the length of each link is commonly $\sqrt{3}\beta$, the length of the Hamiltonian cycle is $\sqrt{3}\beta i$. Since the flight distance is equal to the length of the Hamiltonian cycle, suboptimal scheme $\sigma_{i}^{\left[\Phi_{i}\right]}$ yields the minimum flight distance among the schemes that use tessellations that can enclose the sensor field of radius $\alpha \in (Q_{i-1}^{\left[\Phi_{i-1}\right]},Q_{i}^{\left[\Phi_{i}\right]}]$, i.e., suboptimal scheme $\sigma_{i}^{\left[\Phi_{i}\right]}$ is identical to optimal scheme $\sigma_{\Delta_{\alpha }}^{\left[\Xi_{\alpha }\right]}$. □

For size i∈N, we can find a suboptimal scheme $\sigma_{i}^{\left[\Phi_{i}\right]}$ by completely searching all $\nu_{i}$ patterns and then calculating maximum radius $Q_{i}^{\left[p\right]}$ for each $p\in \{1,\cdots ,\nu_{i}\}$. In general, however, it is hard to obtain suboptimal scheme $\sigma_{i}^{\left[\Phi_{i}\right]}$ since number patterns $\nu_{i}$ increases drastically as size i increases and hence it is hardly tractable to identify all the patterns. Only for relatively small i, we can discover all $\nu_{i}$ patterns, derive maximum radius derive $Q_{i}^{\left[p\right]}$ for each pattern $p\in \{1,\cdots ,\nu_{i}\}$ and then find suboptimal pattern $\Phi_{i}$ and suboptimal scheme $\sigma_{i}^{\left[\Phi_{i}\right]}$ as well. The table below lists suboptimal scheme $\sigma_{i}^{\left[\Phi_{i}\right]}$ for some small i. Also, the table reveals maximum radius $Q_{i}^{\left[\Phi_{i}\right]}$ and flight distance $D\left(\alpha ,\sigma_{i}^{\left[\Phi_{i}\right]}\right)$ achieved by the adoption of scheme $\sigma_{i}^{\left[\Phi_{i}\right]}$ at $\alpha \in (Q_{i-1}^{\left[\Phi_{i-1}\right]},Q_{i}^{\left[\Phi_{i}\right]}]$.

Tessellation $S_{i}^{\left[\Phi_{i}\right]}$, which is employed by suboptimal scheme $\sigma_{i}^{\left[\Phi_{i}\right]}$ in Table 1, is illustrated in Appendix F. Note that the graph induced by tessellation $S_{i}^{\left[\Phi_{i}\right]}$, which is adopted by suboptimal scheme $\sigma_{i}^{\left[\Phi_{i}\right]}$ listed in Table 1, is Hamiltonian. Thus, from Theorem 5, we confirm that suboptimal scheme $\sigma_{i}^{\left[\Phi_{i}\right]}$ is also optimal scheme s$\sigma_{\Delta_{\alpha }}^{\left[\Xi_{\alpha }\right]}$ for radius $\in (Q_{i-1}^{\left[\Phi_{i-1}\right]},Q_{i}^{\left[\Phi_{i}\right]}]$.

Thirdly, a lower bound on minimum flight distance $min\left\{D\left(\alpha ,\sigma_{i}^{\left[p\right]}\right):p\in \left\{1,\cdots ,\nu_{i}\right\},i\in N\right\}$ can be yielded as follows. Recall that each coverage takes the shape of a circular disk with radius β. Then, the apothem of a hexagonal tile inscribed in a coverage is $\frac{\sqrt{3}}{2}\beta$ and consequently its area is equal to $\frac{3\sqrt{3}}{2}\beta^{2}$ (See Figure 8). The area of the sensor field, which takes the shape of a circular disk with radius α, is equal to $\pi \alpha^{2}$. Note that the area of tessellation $S_{i}^{\left[p\right]}$, which is used by scheme $\sigma_{i}^{\left[p\right]}$, must be an integral multiple of the area of a hexagonal tile. Furthermore, the area of $S_{i}^{\left[p\right]}$ should be greater than or equal to the area of the sensor field. Thus, the number of points of interest established by scheme $\sigma_{i}^{\left[p\right]}$ is bounded below as follows.
(12)$$N\left(\alpha ,\sigma_{i}^{\left[p\right]}\right)\geq \left\lceil \frac{2\sqrt{3}\pi }{9}{\left(\frac{\alpha }{\beta }\right)}^{2}\right\rceil$$
for all $p\in \{1,\cdots ,\nu_{i}\}$ and i∈N. Consider a hexagonal tessellation, which consists of $\left\lceil \frac{2\sqrt{3}\pi }{9}{\left(\frac{\alpha }{\beta }\right)}^{2}\right\rceil$ tiles. Assume that the graph induced by the hexagonal tessellation is Hamiltonian. Then, the minimum flight distance is equal to the length of the Hamiltonian cycle multiplied by the distance between two adjacent points of interest, which is $\sqrt{3}\beta \left\lceil \frac{2\sqrt{3}\pi }{9}{\left(\frac{\alpha }{\beta }\right)}^{2}\right\rceil$. Thus, the flight distance achieved by scheme $\sigma_{i}^{\left[p\right]}$ is bounded below by
(13)$$D\left(\alpha ,\sigma_{i}^{\left[p\right]}\right)\geq \sqrt{3}\beta \left\lceil \frac{2\sqrt{3}\pi }{9}{\left(\frac{\alpha }{\beta }\right)}^{2}\right\rceil$$
for all $p\in \{1,\cdots ,\nu_{i}\}$ and i∈N.

## 5. Performance Evaluation of Proposed Scheme

In this section, we evaluate proposed scheme $\sigma^{*}$ by comparing proposed scheme $\sigma^{*}$ and optimal scheme $\sigma^{†}$ in flight distance. Also, we compare the flight distance achieved by proposed scheme $\sigma^{*}$ with the lower bound expressed in (13).

Figure 14 shows the maximum radius (normalized by β) of the sensor field that can be attained by proposed scheme $\sigma^{*}$ with respect to the number of points of interest established by $\sigma^{*}$. In addition, Figure 14 exhibits the maximum radius of sensor field that can be achieved by optimal scheme $\sigma^{†}$. When the number of points of interest is small, the difference between the maximum radii yielded by proposed scheme $\sigma^{*}$ and optimal scheme $\sigma^{†}$ seems to be insignificant.

Figure 15 shows the numbers of points of interest established by proposed scheme $\sigma^{*}$ and optimal scheme $\sigma^{†}$ with respect to the radius of the sensor field (normalized by β. Also, Figure 16 exhibits the flight distances (normalized by β) yielded by proposed scheme $\sigma^{*}$ and optimal scheme $\sigma^{†}$ with respect to the radius (normalized by β) of the sensor field. When the radius of the sensor field is small, the difference between the flight distances attained by proposed scheme $\sigma^{*}$ and optimal scheme $\sigma^{†}$ seems not to be significant. To quantitatively evaluate proposed scheme $\sigma^{*}$ further, define the average difference in flight distance, denoted by $\Delta (\sigma^{*},\sigma^{†},\omega )$, as follows.
(14)$$\Delta \left(\sigma^{*},\sigma^{†},\omega \right)=\frac{1}{\omega }\int_{0}^{\omega }\left|D\left(\alpha ,\sigma^{*}\right)-D\left(\alpha ,\sigma^{†}\right)\right|d\alpha$$
for ω∈(0,∞). Note that optimal scheme $\sigma^{†}$ is available, roughly, at α∈[0,30β]. When ω=30β, average difference $\Delta \left(\sigma^{*},\sigma^{†},\omega \right)$ is calculated to be 0.588β, which is less than 60% of the radius of a coverage. Moreover, the average difference in flight distance is expected to dwindle as observation time ω increases since optimal scheme $\sigma^{†}$ in the sense of minimizing the number of points of interest does not minimize the flight distance more frequently as ω increases.

The table below summarizes specifications of a common multicopter [24,25,26].

Consider a wireless sensor network that employs such a multicopter as a sink node. Suppose that we cover the sensor field of the wireless sensor network with a hexagonal tessellation, establish a point of interest at the centroid of each tile, and construct an air route of the multicopter to visit every point of interest. From Table 2, we obtain the radius of a coverage $\beta =100\;m\times \tan\left(97{}^{\circ}/2\right)\approx 113\;m$. Then, the distance between two adjacent points of interest is equal to $100\;m\times \tan\left(97{}^{\circ}/2\right)\times \sqrt{3}\approx 196\;m$. Let κ denote the number of points of interest established over the sensor field so that the multicopter can finish the round-trip flight without recharging the battery. Then, number κ should necessarily satisfy the following inequality.
(15)$$\kappa \leq \left\lceil \frac{720\;sec\times 5\;m/sec}{100\;m\times \tan\left(97{}^{\circ}/2\right)\times \sqrt{3}}\right\rceil \Leftrightarrow \kappa \leq 18.$$

By employing proposed scheme $\sigma^{*}$, we can cover the sensor field with $T_{1}^{\left(3\right)}$ as far as radius of the sensor field α≤3×β≈339 m from (2). On the other hand, by use of optimal scheme $\sigma^{†}$, we can cover the sensor field with the hexagonal tessellation with 18 tiles, which is mentioned in Table 1 and depicted in Appendix F, if radius α is up to $\sqrt{37}/2\times \beta \approx 344\;m$. Note that a limit on the flight distance is calculated by multiplying the airborne time by the speed of the UAV in (15). In practice, the speed of a UAV is influenced by many factors, e.g., altitude at which the UAV flies and the direction of wind blowing to the UAV. The airborne time highly depends on the battery that operates the UAV. The battery duration is affected by the temperature. The vertical movement of a UAV consumes more energy than horizontal movement. The expression for the maximum number of points of interest in (15) does not include all the influencing factors mentioned above. Nonetheless, it is valid and useful as a rule of thumb.

Consider two cases; Case 1, in which 4 sensor nodes are sparsely scattered in an area as shown in Figure 17a, and Case 2, in which 36 sensor nodes are densely populated in the same as shown in Figure 18a. According to proposed scheme $\sigma^{*}$, tessellation $T_{2}^{\left(1\right)}$ is employed to enclose the sensor field, which takes the shape of a circular disk in which all the sensor nodes are inscribed in both cases. Since tessellation $T_{2}^{\left(1\right)}$ consists of 19 tiles. Nineteen points of interest are established over the sensor field. Note that graph $G_{2}^{\left(1\right)}$, which is induced by tessellation $T_{2}^{\left(1\right)}$, is Hamiltonian. Thus, from Theorem 4, the flight distance of the air route which is constructed along a Hamiltonian cycle in $G_{2}^{\left(1\right)}$ is equal to $\sqrt{3}\times 19\times \beta \approx 32.9\times \beta$ in both cases. (See Figure 17b and Figure 18b). For a comparative evaluation of proposed scheme $\sigma^{*}$, we consider three other methods for establishing points of interest and constructing a round-trip air route that visit every point of interest; Method 1, which establishes a point of interest on each sensor node and constructs a round-trip air route by solving a traveling salesman problem; Method 2, which was introduced in [13]; and Method 3, which was mentioned in [14]. Figure 17c and Figure 18c illustrate the air routes constructed by Method 1 in sparsely and densely populated cases, respectively. From these figures, we obtain the flight distances of the air routes to be $14\sqrt{2}\times \beta \approx 19.8\times \beta$ and 36$\sqrt{2}\times \beta \approx 50.9\times \beta$, respectively, in Cases 1 and 2. Figure 17d and Figure 18d exhibit the air routes constructed by Method 2 in Cases 1 and 2, respectively. From these figures, we observe that two air routes are identical in both cases, and the flight distance of these air routes is equal to 32 × β. Figure 17e demonstrates the air route yielded by Method 3, which is a variant of Method 1. As shown in Figure 17e, the flight distance of the air route is equal to $10\sqrt{2}\times \beta \approx 14.1\times \beta$. From the results above, we can conclude that either Method 1 or Method 2 does not significantly outperform proposed scheme $\sigma^{*}$. In addition to the comparison of flight distances, we should consider the following factors in the comparative evaluation. First, to use Method 1 (or Method 4), points of interest should be bijectively established on (or near) sensor nodes. Thus, precise locational information about sensor nodes should be available. On the other hand, proposed scheme $\sigma^{*}$ can be applied only with vague information about the geographic distribution of sensor nodes. Secondly, the UAV always receive data from a single sensor node when Method 1 or Method 3 is used. On the other hand, the UAV occasionally has to receive data from two or more sensor nodes if the proposed scheme is employed. Thus, a scheduling-based or contention-based medium access control scheme is needed, which brings about the consumption of additional time resources for signaling and throughput degradation. Thirdly, the area in which sensor nodes lie is divided into a number of regions in order to use Method 2. Then, a region takes the shape of a square in which a coverage is inscribed. Consequently, there can be a sensor node that is never able to deliver data to the UAV at whichever point of interest the UAV is. Fourthly, the distribution of sensor nodes can be changed as time goes on. As implied in Figure 17c and Figure 18c, Method 1, which places a point of interest at each sensor node, is highly sensitive to a change occurring in the number of sensor nodes and their locations. On the other hand, the proposed scheme is relatively insensitive to such a change, even though the throughput from sensor nodes to the UAV is affected to some extent.

## 6. Conclusions

In this paper, we considered a wireless sensor network that employs a UAV as a sink node. The wireless sensor network under discussion also lies in a harsh environment, so the geographical distribution of sensor nodes is barely known to us. In the network, the UAV flies along a round-trip air route that visits every point of interest. Whenever the UAV arrives at a point of interest, the UAV hovers and gathers data from neighboring sensor nodes on the ground. To satisfy the requirement that the UAV should be able to receive data from all the sensor nodes and to follow the recommendation that a shorter round-trip air route of the UAV is more preferred as well, we proposed a scheme characterized by covering the sensor field of the wireless sensor network with three classes of hexagonal tessellations, establishing a point of interest at the centroid of each tile belonging to the adopted tessellation, and constructing an air route of the UAV that visits every point of interest along a Hamiltonian cycle on the graph induced by the employed tessellation. Next, we developed a closed-form expression for the exact flight distance attained by the proposed scheme. For comparative evaluation, we discovered an optimal scheme that minimizes the flight distance for a given radius of the sensor field by completely inspecting patterns and corroborating the property of Hamiltonicity. Also, we presented a universal lower bound on the flight distance. Numerical examples showed that the flight distance attained by the proposed scheme is only slightly longer compared with the flight distance yielded by an optimal scheme. Furthermore, the proposed scheme was proven to be practically valid when a common multicopter is employed as a sink node.

## Figures and Tables

**Figure 1 sensors-24-03867-f001:**
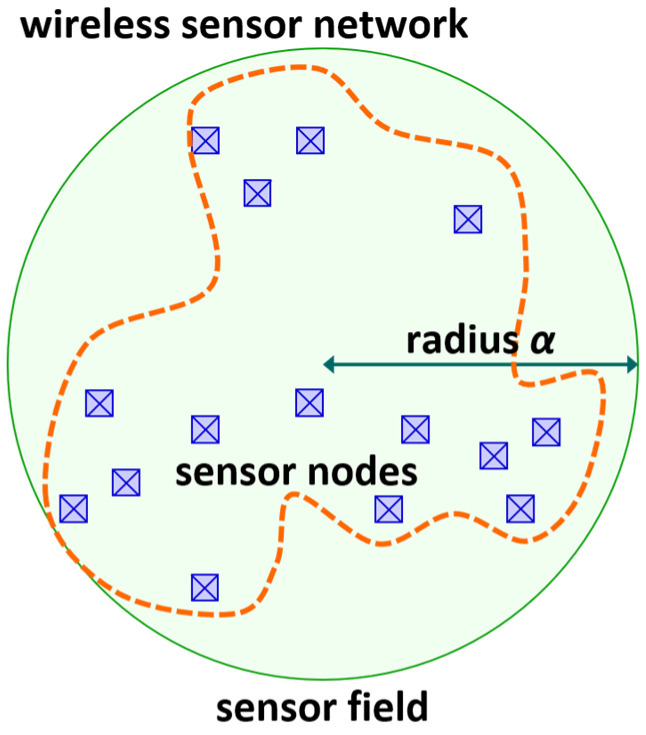
Sensor field of wireless sensor network which is in the shape of circular disk.

**Figure 2 sensors-24-03867-f002:**
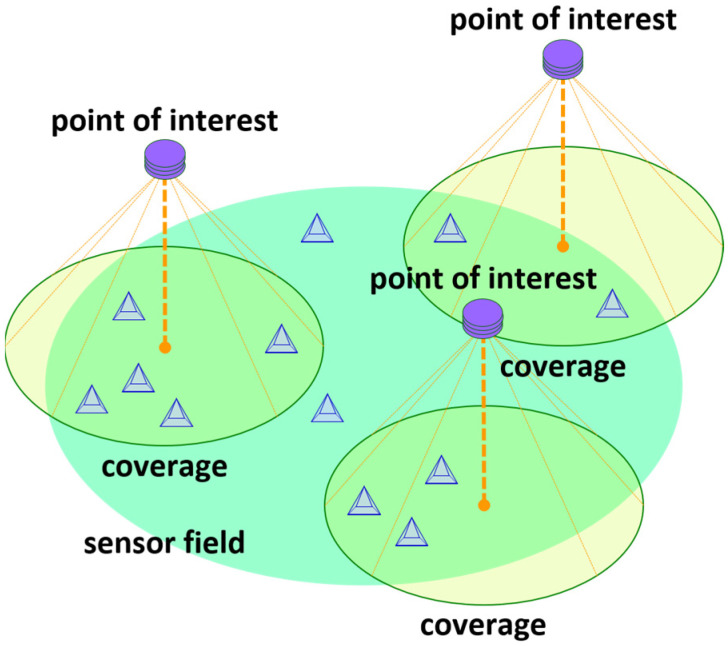
Points of interest established over sensor field of wireless sensor network.

**Figure 3 sensors-24-03867-f003:**
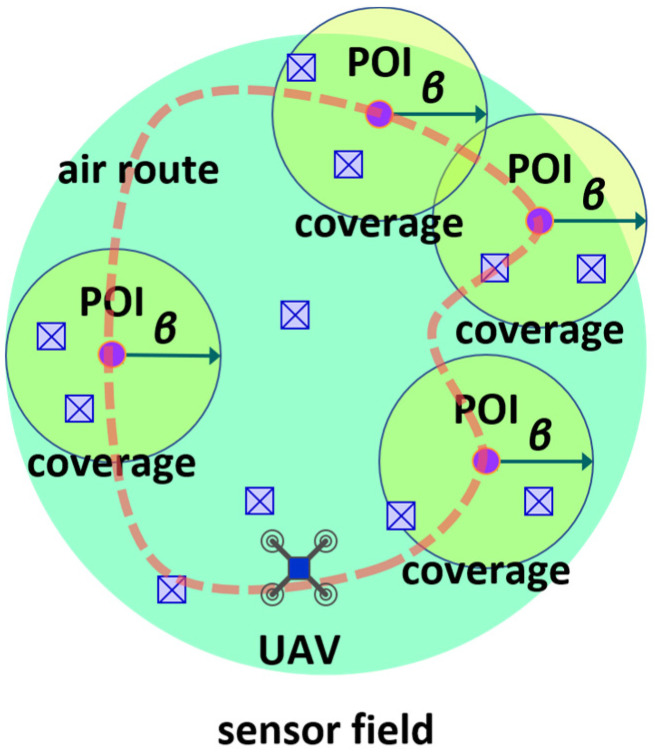
Coverage associated with point of interest which is in the shape of circular disk.

**Figure 4 sensors-24-03867-f004:**
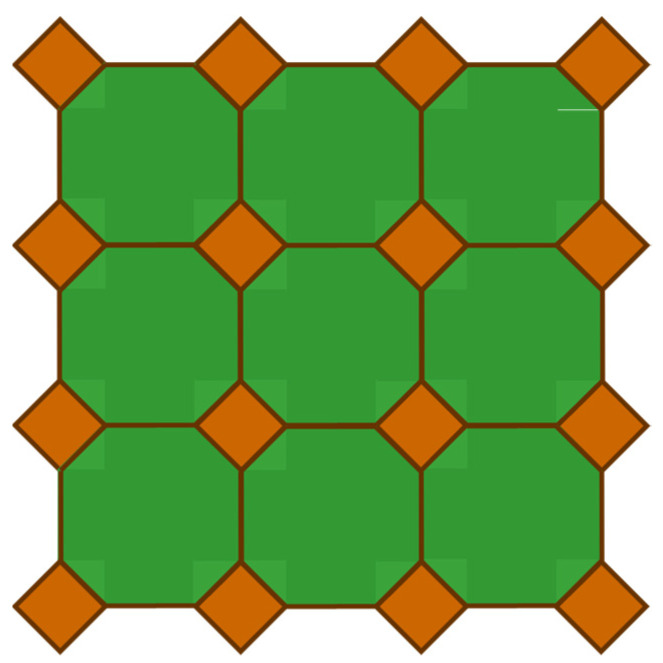
Tessellation consisting of tiles in two polygonal shapes.

**Figure 5 sensors-24-03867-f005:**
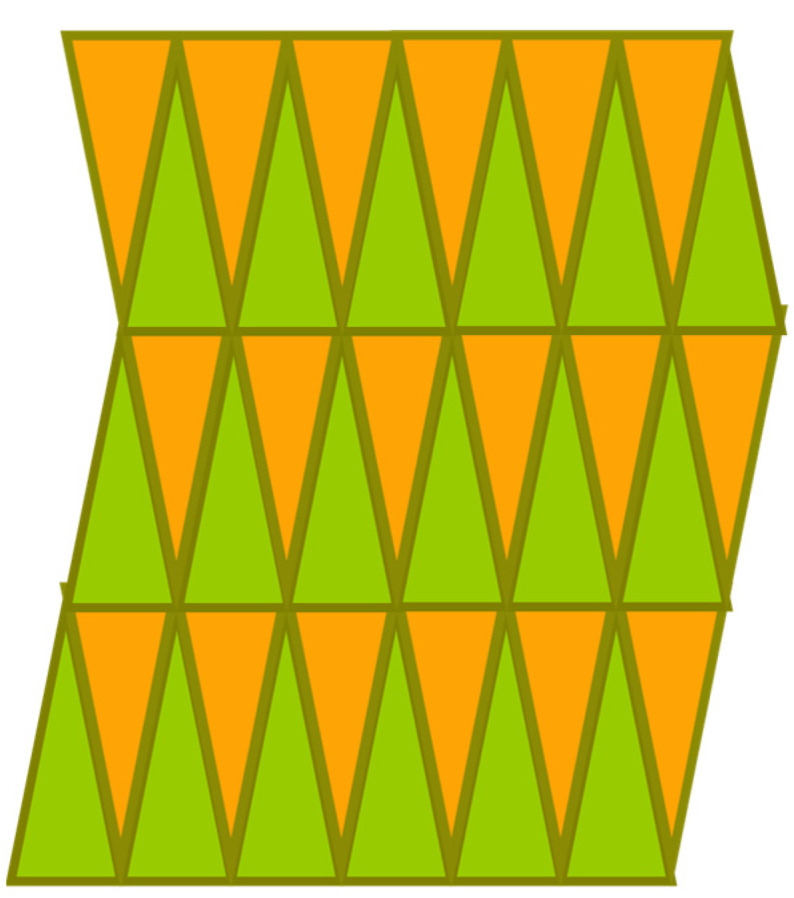
Monohedral tessellation consisting of triangular tiles.

**Figure 6 sensors-24-03867-f006:**
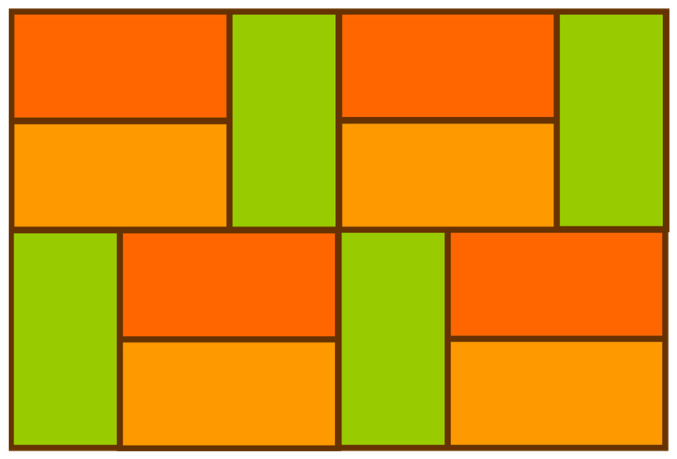
Non-edge-to-edge monohedral tessellation consisting of rectangular tiles.

**Figure 7 sensors-24-03867-f007:**
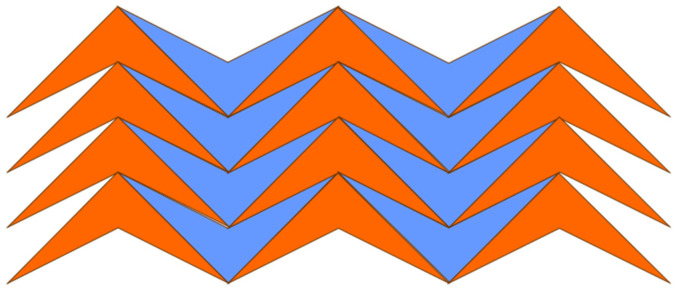
Edge-to-edge monohedral tessellation comprised by tiles that take the shape of a concave polygon.

**Figure 8 sensors-24-03867-f008:**
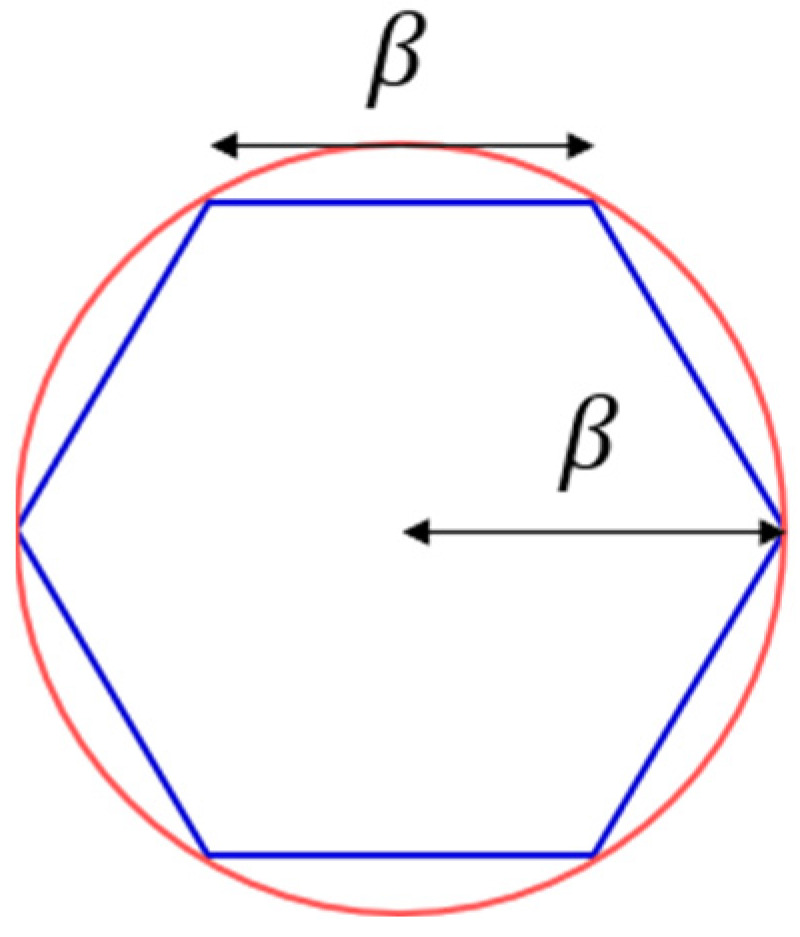
Hexagonal tile inscribed in coverage taking shape of circular disk with radius β.

**Figure 9 sensors-24-03867-f009:**
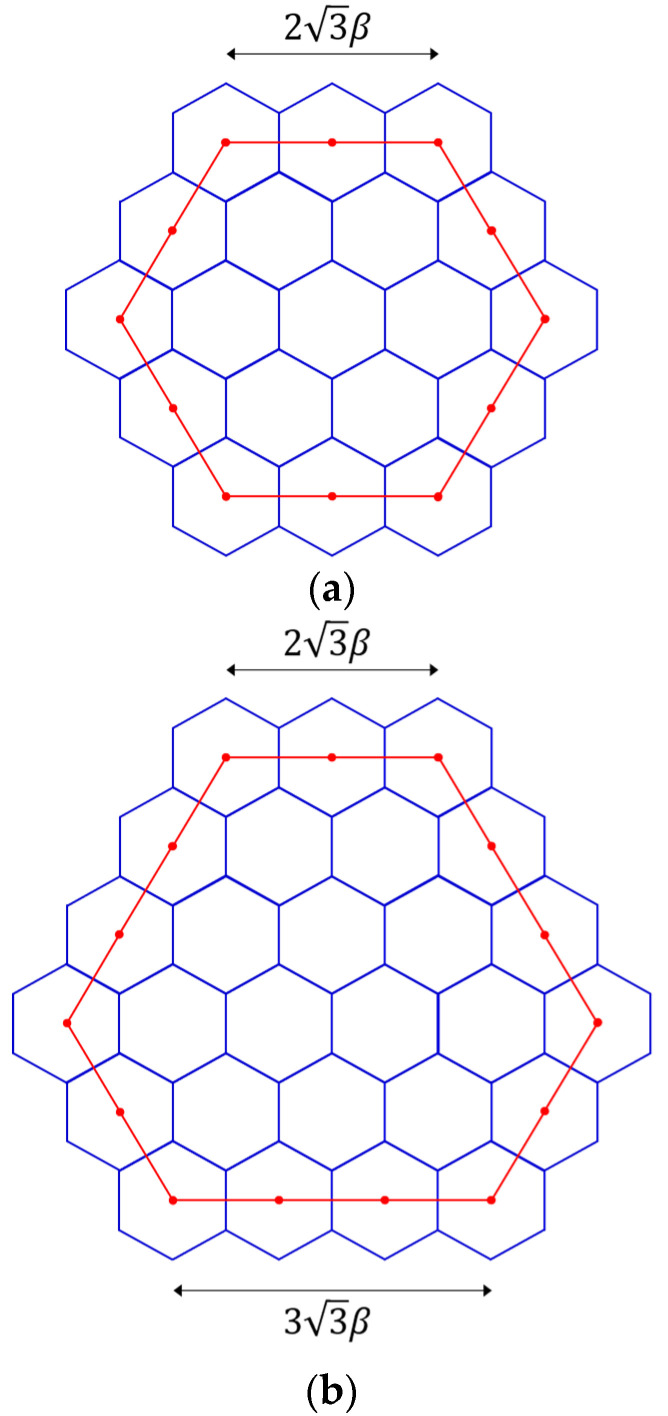
Three categories of hexagonal tessellations. (**a**) Tessellation $T_{2}^{\left(1\right)}$; (**b**) Tessellation $T_{2}^{\left(2\right)}$; (**c**) Tessellation $T_{2}^{\left(3\right)}$.

**Figure 10 sensors-24-03867-f010:**
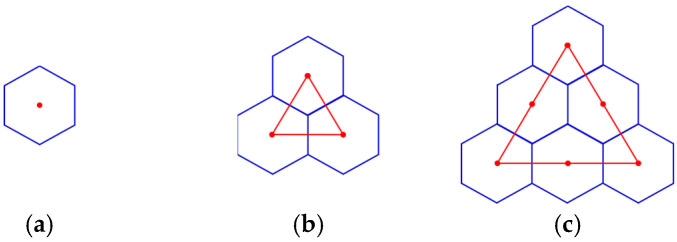
Tessellations $T_{0}^{\left(1\right)}$, $T_{0}^{\left(2\right)}$ and $T_{0}^{\left(3\right)}$. (**a**) Tessellation $T_{0}^{\left(1\right)}$; (**b**) Tessellation $T_{0}^{\left(2\right)}$; (**c**) Tessellation $T_{0}^{\left(3\right)}$.

**Figure 11 sensors-24-03867-f011:**
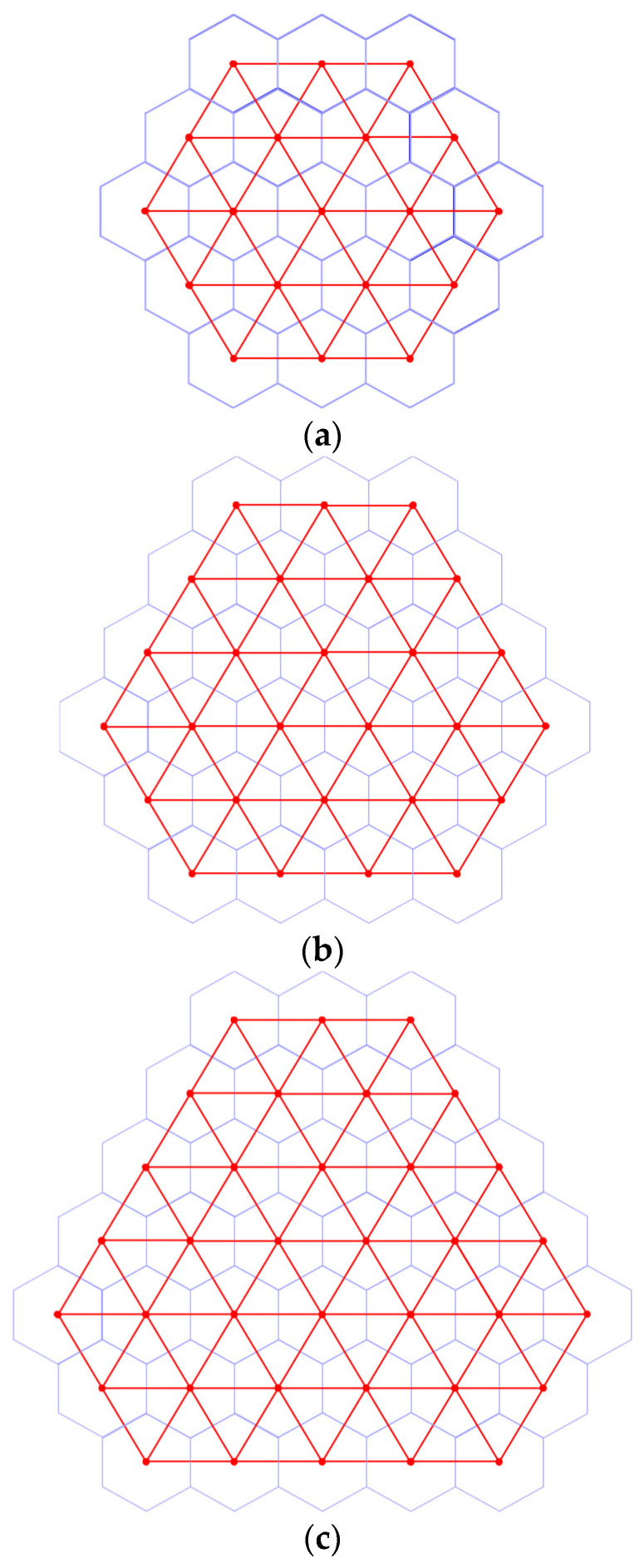
Graphs $G_{2}^{\left(1\right)}$, $G_{2}^{\left(2\right)}$ and $G_{2}^{\left(3\right)}$ induced by tessellations $T_{2}^{\left(1\right)}$, $T_{2}^{\left(2\right)}$ and $T_{2}^{\left(3\right)}$. (**a**) Graph $G_{2}^{\left(1\right)}$ induced by tessellation $T_{2}^{\left(1\right)}$; (**b**) Graph $G_{2}^{\left(2\right)}$ induced by tessellation $T_{2}^{\left(2\right)}$; (**c**) Graph $G_{2}^{\left(3\right)}$ induced by tessellation $T_{2}^{\left(3\right)}$.

**Figure 12 sensors-24-03867-f012:**
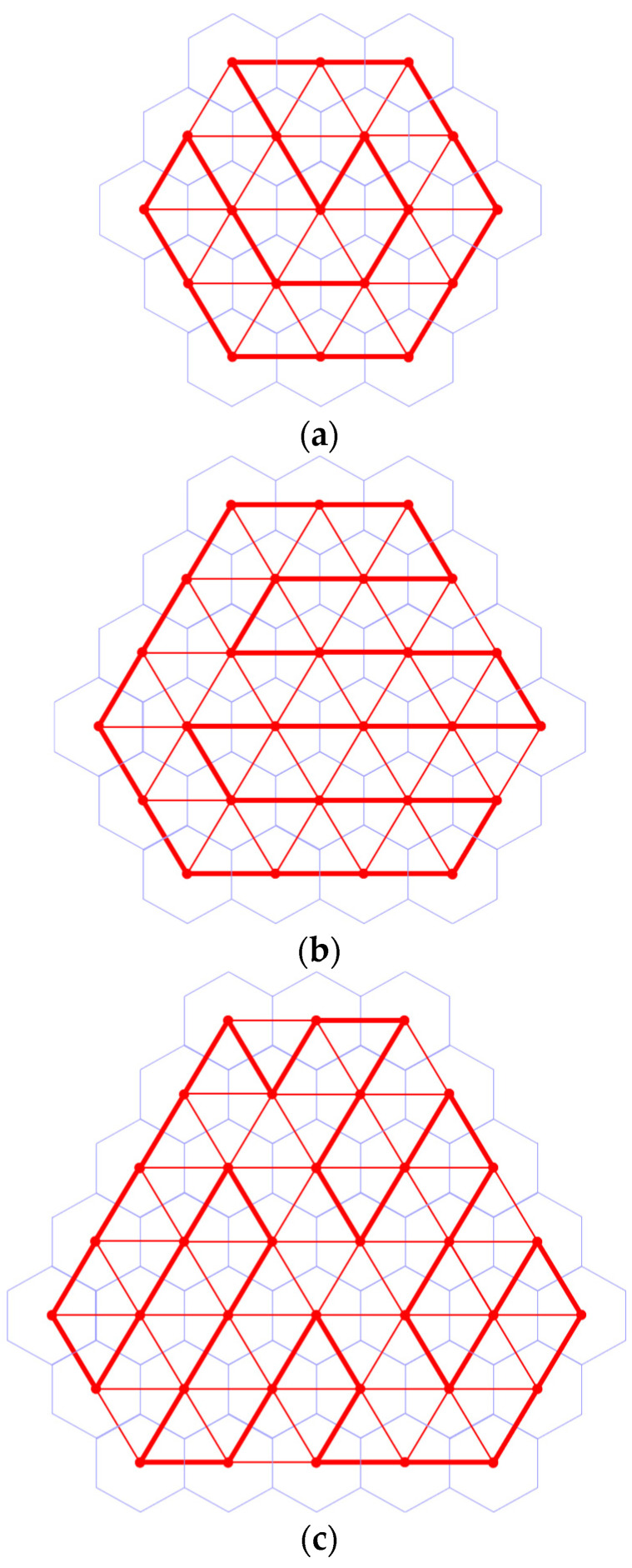
Hamiltonian cycles in graphs $G_{2}^{\left(1\right)}$, $G_{2}^{\left(2\right)}$ and $G_{2}^{\left(3\right)}$. (**a**) Hamiltonian cycle in graph $G_{2}^{\left(1\right)}$; (**b**) Hamiltonian cycle in graph $G_{2}^{\left(2\right)}$; (**c**) Hamiltonian cycle in graph $G_{2}^{\left(3\right)}$.

**Figure 13 sensors-24-03867-f013:**
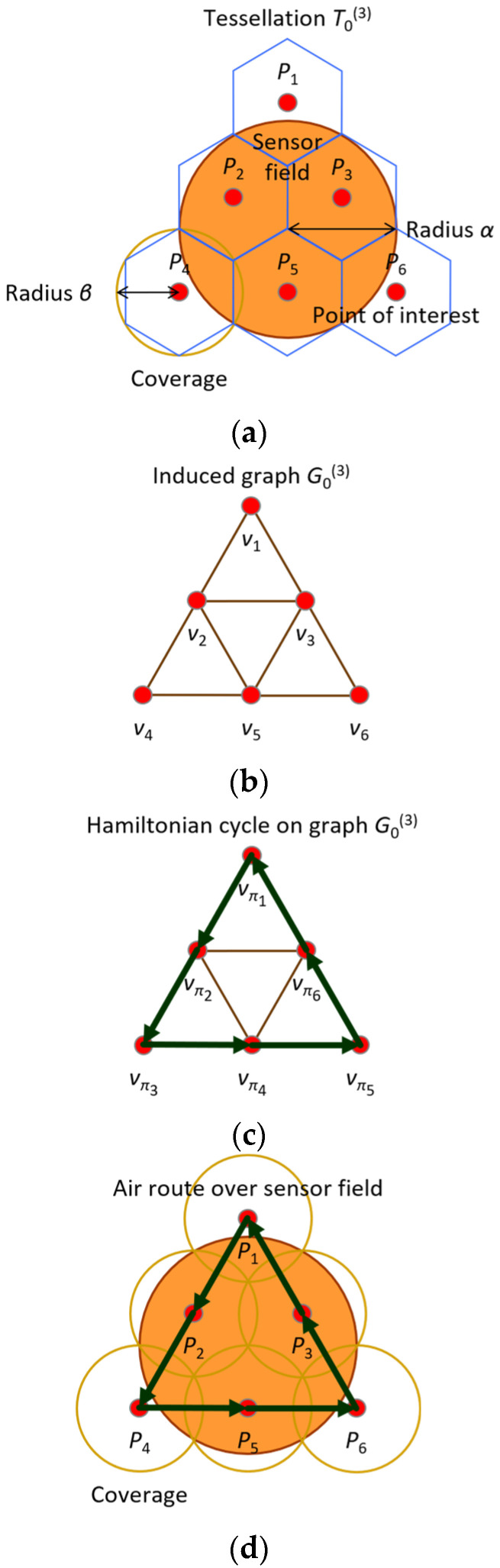
Exemplary establishment of points of interest and construction of round-trip air route according to proposed scheme $\sigma^{*}$. (**a**) Enclosement of sensor filed with tessellation $T_{0}^{\left(3\right)}$ and establishment of points of interest denoted by $P_{1},\cdots ,P_{6}$; (**b**) inducement of graph $G_{0}^{\left(3\right)}$ from tessellation $T_{0}^{\left(3\right)}$; (**c**) discovery of Hamiltonian cycle $\left\{v_{1},v_{2},v_{4},v_{5},v_{6},v_{3},v_{1}\right\}$ in graph $G_{0}^{\left(3\right)}$; (**d**) construction of air route $\left\{P_{1},P_{2},P_{4},P_{5},P_{6},P_{3},P_{1}\right\}$ over sensor field.

**Figure 14 sensors-24-03867-f014:**
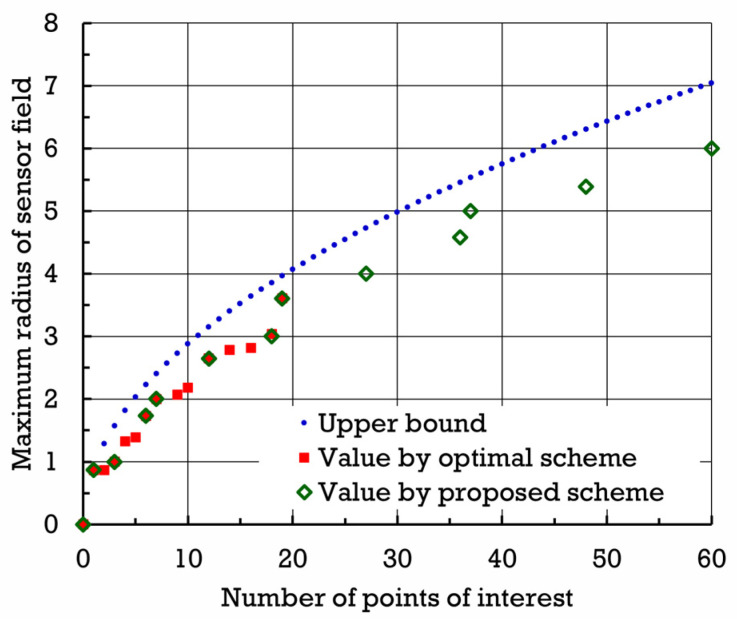
Maximum radii (normalized by β) of sensor field yielded by proposed scheme $\sigma^{*}$ and optimal scheme $\sigma^{†}$ with respect to number of points of interest established over sensor field.

**Figure 15 sensors-24-03867-f015:**
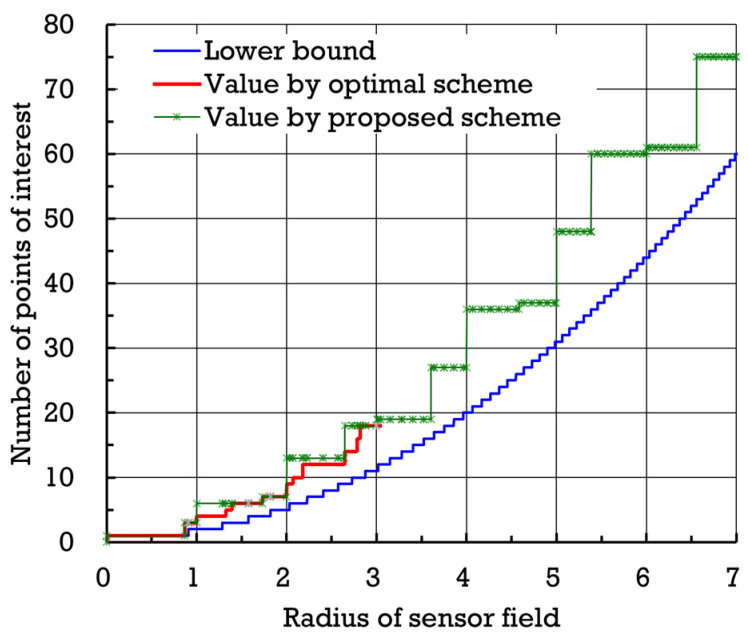
Numbers of points of interest established by proposed scheme $\sigma^{*}$ and optimal scheme $\sigma^{†}$ with respect to the radius of the sensor field.

**Figure 16 sensors-24-03867-f016:**
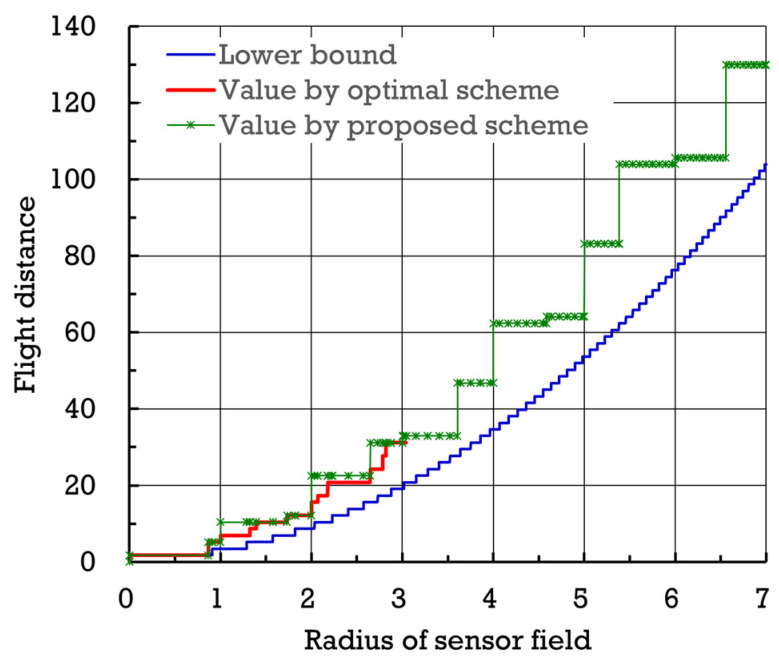
Flight distances (normalized by β) yielded by proposed scheme $\sigma^{*}$ and optimal scheme $\sigma^{†}$ with respect to the radius (normalized by β) of the sensor field.

**Figure 17 sensors-24-03867-f017:**
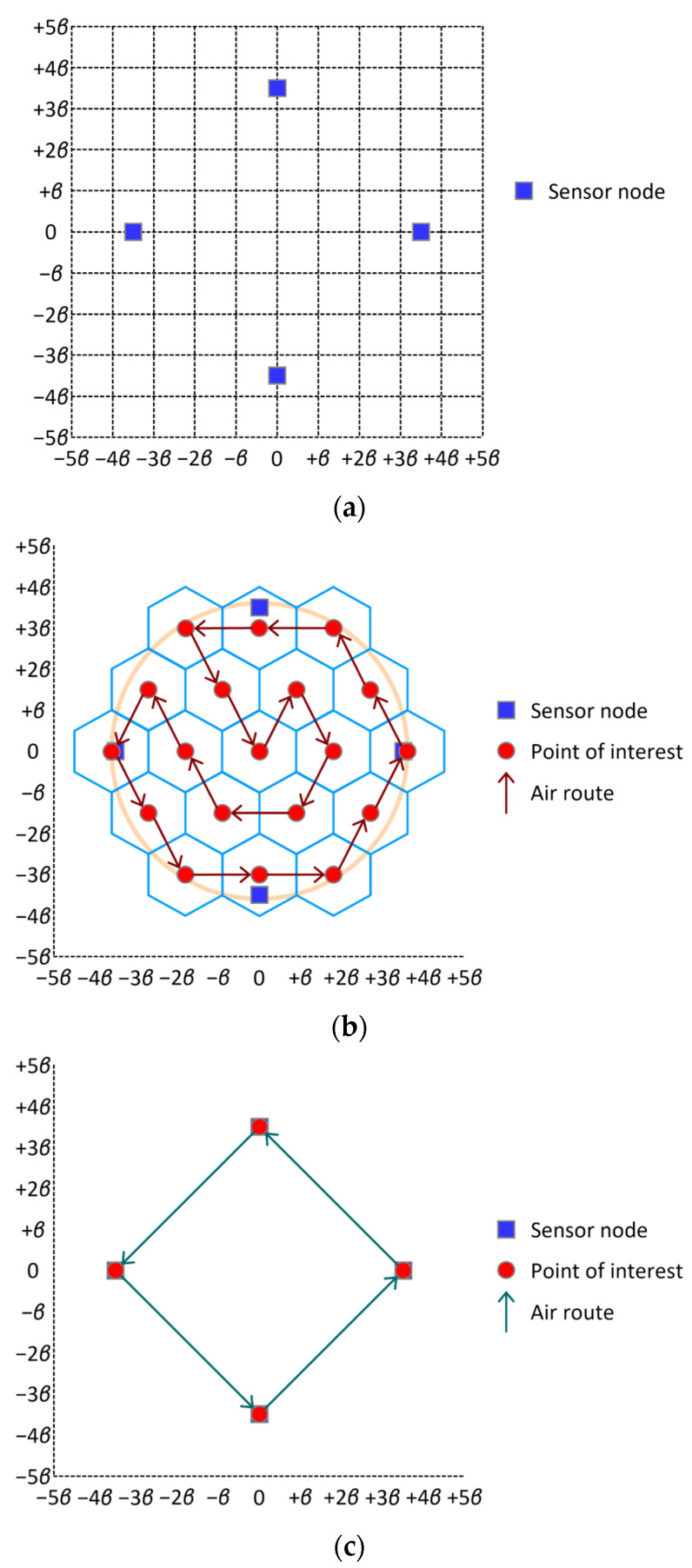
Comparison of flight distances of air routes constructed by proposed scheme $\sigma^{*}$ and Methods 1, 2, and 3. (**a**) Sparsely scattered sensor nodes on the ground; (**b**) air route constructed by proposed scheme $\sigma^{*}$; (**c**) air route constructed by Method 1; (**d**) air route constructed by Method 2; (**e**) air route constructed by Method 3.

**Figure 18 sensors-24-03867-f018:**
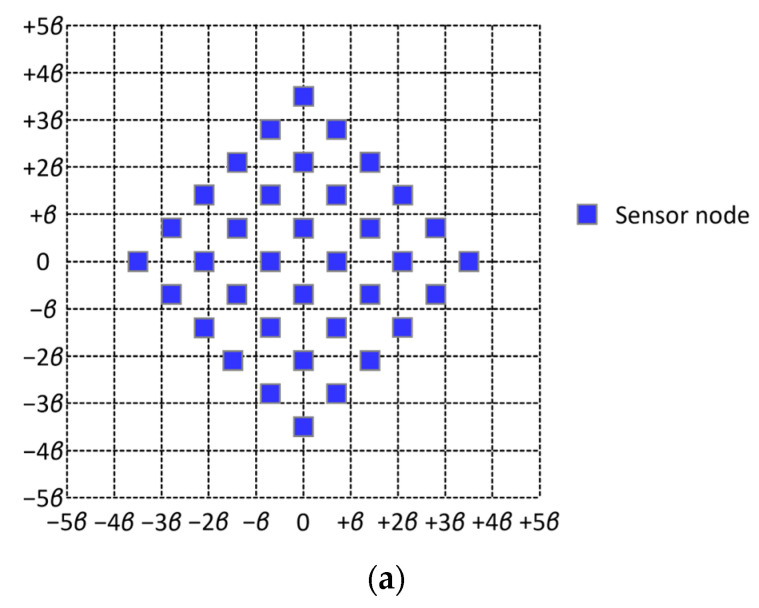
Comparison of flight distances of air routes constructed by proposed scheme $\sigma^{*}$ and Methods 1, 2, and 3. (**a**) Densely populated sensor nodes on the ground; (**b**) air route constructed by proposed scheme $\sigma^{*}$; (**c**) air route constructed by Method 1; (**d**) air route constructed by Method 2.

**Table 1 sensors-24-03867-t001:** Scheme $\sigma_{i}^{\left[\Phi_{i}\right]}$, maximum radius $Q_{i}^{\left[\Phi_{i}\right]}$, and flight distance $D\left(\alpha ,\sigma_{i}^{\left[\Phi_{i}\right]}\right)$.

Scheme $\sigma_{i}^{\left[\Phi_{i}\right]}$	Maximum Radius $Q_{i}^{\left[\Phi_{i}\right]}$	$\mathrm{Is}\;\mathrm{Graph}\;\mathrm{Induced}\;\mathrm{by}\;S_{i}^{\left[\Phi_{i}\right]}$ Hamiltonian?	Flight Distance $D\left(\alpha ,\sigma_{i}^{\left[\Phi_{i}\right]}\right)$
$\sigma_{1}^{\left[\Phi_{1}\right]}$	$\sqrt{3}/2\cdot \beta$	Yes	$\sqrt{3}\cdot \beta$
$\sigma_{3}^{\left[\Phi_{3}\right]}$	β	Yes	$3\sqrt{3}\cdot \beta$
$\sigma_{4}^{\left[\Phi_{4}\right]}$	$\sqrt{7}/2\cdot \beta$	Yes	$4\sqrt{3}\cdot \beta$
$\sigma_{5}^{\left[\Phi_{5}\right]}$	$(6\sqrt{3}-9)\cdot \beta$	Yes	$5\sqrt{3}\cdot \beta$
$\sigma_{6}^{\left[\Phi_{6}\right]}$	$\sqrt{3}\cdot \beta$	Yes	$6\sqrt{3}\cdot \beta$
$\sigma_{7}^{\left[\Phi_{7}\right]}$	2·β	Yes	$7\sqrt{3}\cdot \beta$
$\sigma_{9}^{\left[\Phi_{9}\right]}$	$43\sqrt{3}/36\cdot \beta$	Yes	$9\sqrt{3}\cdot \beta$
$\sigma_{10}^{\left[\Phi_{10}\right]}$	$\sqrt{19}/2\cdot \beta$	Yes	$10\sqrt{3}\cdot \beta$
$\sigma_{12}^{\left[\Phi_{12}\right]}$	$\sqrt{7}\cdot \beta$	Yes	$12\sqrt{3}\cdot \beta$
$\sigma_{14}^{\left[\Phi_{14}\right]}$	$\sqrt{31}/2\cdot \beta$	Yes	$14\sqrt{3}\cdot \beta$
$\sigma_{16}^{\left[\Phi_{16}\right]}$	31/11·β	Yes	$16\sqrt{3}\cdot \beta$
$\sigma_{18}^{\left[\Phi_{18}\right]}$	$\sqrt{37}/2\cdot \beta$	Yes	$18\sqrt{3}\cdot \beta$

**Table 2 sensors-24-03867-t002:** Specifications of multicopter.

Parameter	Unit	Value
Altitude	Meters	100
Speed	Meters/second	5
Beamwidth of receiving antenna	Degrees	97
Airborne time	Seconds	720

## Data Availability

Data are contained within the article.

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
