# Peer review of "Tessellation-Based Construction of Air Route for Wireless Sensor Networks Employing UAV"

_sensors, 2024, doi:10.3390/s24123867_

Round 1

Reviewer 1 Report

Comments and Suggestions for Authors

The paper is focused on the effective method of an UAV route selection over the sensors that can wirelessly  send the data to an UAV in specified point of interests. The proposal is interesting and clearly described in the paper. I do not see any critical comments. The proposal has been evaluated and its performance is discussed in Section 5. Of course I understand that these are theoretical investigations with ideal assumptions, which should be verified using real conditions  (more consistent with real UAV, wireless transmission and terrain profile characteristics).

Author Response

Dear reviewer

We would like to thank you for your valuable and constructive comments. Upon your comments, we carefully revised our paper and made a response.

Reviewer 2 Report

Comments and Suggestions for Authors

In this article authors propose a scheme for UAV-based sensor data collection in wireless sensor networks, in which the sensor field is covered by  using 3 different classes of hexagonal tessellations, the points of interest are placed in each tessellation and an air route of the UAV which visits every point of interest along a Hamiltonian cycle on the graph defined by the employed tessellation is established. The closed-form expression for the exact flight distance for the proposed scheme is derived, and compared to the heuristically found the flight distance of found 'optimal' scheme with the minimal number of points of interest.

The paper is well written and the given subject is addequately presented. However, some minor issues should be corrected:

- The claimed contribution should be defined more strict and clear manner. The current version in lined 99-114 is the list of things done, and not the specific contribution which must be supporeted by the analysis and the results.

- In the introduction section some references to the previous work is given. However, the comparison or even discussion of here proposed scheme and these previous ones is not given.

- The optimal solutions is the best that exists i.e. optimal. It is not clear how you can optian optimal solution heuristically - by definition this is a suboptimal solution. Please reconsider the use of terms optimal and revise the descriptions and discussion related to this issue.

Comments on the Quality of English Language

The Quality of English Language is acceptable.

Author Response

(The authors gave the same response as above.)

Reviewer 3 Report

Comments and Suggestions for Authors

1. The paper heavily relies on theoretical analysis and mathematical modeling, with limited empirical validation or experimental results to support the proposed scheme. In addition, trajectory planning is also a key for employing UAV for collecting sensor data. Authors may discuss more, for example: Adaptive Multi-UAV Trajectory Planning Leveraging Digital Twin Technology for Urban IIoT Applications, IEEE Transactions on Network Science and Engineering

2. The assumptions made in the paper regarding the circular sensor field and identical coverage areas for each point of interest may limit the applicability of the proposed scheme to more complex and realistic sensor network deployments. Relaxing these assumptions could enhance the robustness and generalizability of the scheme.

3. The proposed scheme's performance is compared to an optimal scheme that minimizes the number of points of interest, but it is not compared to other baseline schemes or existing solutions in the literature. Adding comparative analysis with other schemes would strengthen the evaluation.

4. The paper does not discuss the communication overhead or energy consumption implications of the proposed scheme. Analyzing these aspects would provide a more comprehensive evaluation of the scheme's efficiency and scalability.

5. The paper focuses solely on the flight path optimization of the UAV sink, without considering other factors such as task scheduling, data processing, or collaboration between multiple UAVs. Addressing these aspects could lead to a more integrated and optimized solution.

6. The paper could benefit from a more detailed discussion on how the proposed scheme might be adapted or extended to handle dynamic changes in the sensor network, such as node failures, mobility, or environmental effects.

7. The paper heavily utilizes mathematical notation and symbols, which may pose accessibility challenges for readers not familiar with graph theory or tessellation concepts. Providing more intuitive explanations or graphical illustrations could enhance clarity and readability.

8. The paper could be improved by including a discussion on the practical constraints and limitations of the proposed scheme, such as the maximum number of points of interest that can be effectively visited by a UAV, or the effects of environmental factors like wind.

Comments on the Quality of English Language

NONE

Author Response

(The authors gave the same response as above.)

Round 2

Reviewer 2 Report

Comments and Suggestions for Authors

The paper is well written and the given subject is addequately presented.In the revised manuscript authors answered on the issues reported in reviewer comments, and updated the manuscript accordingly.

Comments on the Quality of English Language

The Quality of English Language is acceptable.